# Early Treatment Outcomes for Bloodstream Infections Caused by Potential AmpC Beta-Lactamase-Producing Enterobacterales with Focus on Piperacillin/Tazobactam: A Retrospective Cohort Study

**DOI:** 10.3390/antibiotics10060665

**Published:** 2021-06-02

**Authors:** Lena Herrmann, Aurelia Kimmig, Jürgen Rödel, Stefan Hagel, Norman Rose, Mathias W. Pletz, Christina Bahrs

**Affiliations:** 1 Institute of Infectious Diseases and Infection Control, Jena University Hospital, Am Klinikum 1, 07747 Jena, Germany; lena.herrmann@uni-jena.de (L.H.); aurelia.kimmig@med.uni-jena.de (A.K.); stefan.hagel@med.uni-jena.de (S.H.); norman.rose@med.uni-jena.de (N.R.); mathias.pletz@med.uni-jena.de (M.W.P.); 2Institute of Medical Microbiology, Jena University Hospital, 07747 Jena, Germany; Juergen.Roedel@med.uni-jena.de; 3 Clinic for Anaesthesiology and Intensive Care Medicine, Jena University Hospital, 07747 Jena, Germany; 4Department of Medicine I, Division of Infectious Diseases and Tropical Medicine, Medical University of Vienna, 1090 Vienna, Austria

**Keywords:** bacteremia, AmpC beta-lactamase, *Enterobacter*, *Serratia*, piperacillin/tazobactam

## Abstract

The Gram-negative bacilli *Serratia* spp., *Providencia* spp., *Morganella morganii*, *Citrobacter freundii* complex, *Enterobacter* spp. and *Klebsiella aerogenes* are common Enterobacterales that may harbor inducible chromosomal AmpC beta-lactamase genes. The purpose of the present study was to evaluate treatment outcomes and identify predictors of early treatment response in patients with bloodstream infection caused by potential AmpC beta-lactamase-producing Enterobacterales (SPICE-BSI). This cohort study included adult patients with SPICE-BSI hospitalized between 01/2011 and 02/2019. The primary outcome was early treatment response 72 h after the start of active treatment, defined as survival, hemodynamic stability, improved or stable SOFA score, resolution of fever and leukocytosis and microbiologic resolution. Among 295 included patients, the most common focus was the lower respiratory tract (27.8%), and *Enterobacter* spp. (*n* = 155) was the main pathogen. The early treatment response rate was significantly lower (*p* = 0.006) in the piperacillin/tazobactam group (17/81 patients, 21.0%) than in the carbapenem group (40/82 patients, 48.8%). Independent negative predictors of early treatment response (*p* < 0.02) included initial SOFA score, liver comorbidity and empiric piperacillin/tazobactam treatment. In vitro piperacillin/tazobactam resistance was detected in three patients with relapsed *Enterobacter*-BSI and initial treatment with piperacillin/tazobactam. In conclusion, our findings show that piperacillin/tazobactam might be associated with early treatment failure in patients with SPICE-BSI.

## 1. Introduction

The Gram-negative bacilli *Serratia* spp., *Providencia* spp., *Morganella morganii*, *Citrobacter freundii* complex, *Enterobacter* spp. and *Klebsiella aerogenes* (also named as “SPICE” organisms) are common Enterobacterales that may harbor inducible chromosomal AmpC beta-lactamase genes [1,2]. Often, the AmpC beta-lactamase genes are initially suppressed, but exposure to antibiotics, particularly broad-spectrum cephalosporins, can induce their expression. Hyperproduced/derepressed AmpC beta-lactamase can inactivate third-generation cephalosporins, most penicillins, and some beta-lactam/beta-lactamase inhibitor combinations [3,4]. SPICE organisms can cause a variety of severe, mostly hospital-acquired infections [5,6], including bloodstream infection (BSI), urinary tract infection, pneumonia, biliary and abdominal infection, surgical site infection and device-associated infection [7,8,9]. There is strong agreement that third-generation cephalosporin treatment should be avoided for SPICE infections irrespective of in vitro susceptibility; however, piperacillin/tazobactam treatment as a carbapenem-sparing regimen is still a matter of debate [10,11,12,13].

The purpose of the present study was to evaluate treatment outcomes of the most common empiric antibiotic regimens in hospitalized patients with potential AmpC beta-lactamase-producing Enterobacterales BSI and to identify independent predictors of early treatment response in clinical practice.

## 2. Results

### 2.1. Study Cohort

Over an 8-year study period, we screened a total of 340 adult patients with BSI caused by any SPICE organism for inclusion in the study. Forty-five of whom had to be excluded due to antibiotic therapy of less than 72 h at Jena University Hospital (see Figure 1). 

*Enterobacter* spp. (*n* = 155) were the most common SPICE pathogens, followed by *Serratia* spp. (*n* = 78), *Citrobacter freundii* complex (*n* = 37), *Morganella morganii* (*n* = 14), *Klebsiella aerogenes* (*n* = 9) and *Providencia* spp. (*n* = 6). *Enterobacter* spp. included *Enterobacter cloacae* complex (*n* = 154) and *Enterobacter cancerogenus* (*n* = 1), *Serratia* spp. included *Serratia marcescens* (*n* = 75), *Serratia liquefaciens* (*n* = 2) and *Serratia odorifera* (*n* = 1), *Providencia* spp. included *Providencia rettgeri* (*n* = 3) and *Providencia stuartii* (*n* = 3). In 4 patients (1.4%), two different SPICE species were detected in the initial BC.

Of the 295 patients included, 204 (69.2%) were males, and 91 (30.8%) were females. The mean age of the patients was 65.3 (+/− SD 13.7) years. A total of 109 patients (36.9%) were admitted to an intensive care unit (ICU), and the overall 14-day mortality rate was 7.5%. The most common source of infection was pneumonia (*n* = 82, 27.8%), followed by primary bacteremia (*n* = 65, 22.0%), urinary tract infection (*n* = 53, 18.0%), acute cholangitis (*n* = 37, 12.5%), vascular catheter-related BSI (*n* = 34, 11.5%), surgical site infection (*n* = 27, 9.2%) and intra-abdominal infection (*n* = 16, 5.4%). Less common infections included skin/soft tissue infection (*n* = 8, 2.7%), endocarditis (*n* = 4, 1.4%), mediastinitis (*n* = 4, 1.4%), joint/bone infection (*n* = 2, 0.7%), left ventricular assist device-related infection (*n* = 2, 0.7%) and prostatitis (*n* = 1, 0.3%). The source of bacteremia was microbiologically confirmed in 122 patients (41.4%); 34 patients (11.5%) had more than one source of bacteremia.

### 2.2. Empiric Antimicrobial Treatment

The two most frequently used empiric regimes were carbapenem monotherapy (*n* = 82) and piperacillin/tazobactam monotherapy (*n* = 81). Carbapenem monotherapy involved meropenem (*n* = 80) and imipenem/cilastatin (*n* = 2). Other empiric regimens included fluoroquinolone monotherapy (*n* = 53), any combination therapy (*n* = 38), cephalosporin monotherapy (*n* = 32), cotrimoxazole monotherapy (*n* = 8), and gentamicin monotherapy (*n* = 1). The used fluoroquinolones were ciprofloxacin (*n* = 45) and moxifloxacin (*n* = 8). Cephalosporin monotherapy involved cefuroxime (*n* = 2), ceftriaxone (*n* = 19), cefotaxime (*n* = 1) and ceftazidime (*n* = 10). The median duration of antimicrobial treatment (empiric and definitive therapy) was 11 days (interquartile range: 8.0-14.0 days). The distribution of baseline characteristics, including demographic characteristics, comorbidities, disease severity and BSI source, among the different empiric treatment groups is shown in Table 1. 

Patients receiving combination therapy had the highest severity of disease (median SOFA score: 7.5) and highest rates of cardiac or orthopedic implanted devices (52.6%) and pneumonia as sources of bacteremia (57.9%). Among patients receiving monotherapy, those under empiric carbapenem monotherapy had the highest SOFA scores (median 4.5) and the highest pneumonia rates (35.4%). Patients with empiric piperacillin/tazobactam monotherapy had the lowest rates of urinary tract infection (8.6%). Patients receiving cotrimoxazole had the youngest ages (median age 50.5 years), the lowest severity of disease (median SOFA score: 1.5), and they often suffered from urinary tract infection (37.5%).

Initial antimicrobial treatment was changed in 125/295 patients (42.4%) after a median of 3 days (interquartile range: 3–5 days). The reasons for change included escalation (*n* = 58), de-escalation (*n* = 42), and other (no clear escalation or de-escalation; *n* = 25). The empiric piperacillin/tazobactam monotherapy group had the highest escalation rate (*n* = 30; 37.0%), while the combination therapy group had the highest de-escalation rate (*n* = 25; 65.8%).

### 2.3. Primary Outcome

Overall, early treatment response 72 h after the initial BC and start of active therapy was achieved in 119/295 patients (40.3%) with SPICE-BSI. Only 18/109 ICU patients (16.5%) exhibited early treatment response versus 101/186 normal-ward patients (54.3%). When the two most common empiric regimens were compared, the early treatment response rate was significantly lower (*p* = 0.006) in the piperacillin/tazobactam group (17/81 patients, 21.0%) than in the carbapenem group (40/82 patients, 48.8%). Comparing both regimens under statistical control of covariates (i.e., baseline characteristics, comorbidities, disease severity and bacteremia sources) with absolute standardized mean differences ≥ 0.2 resulted in an adjusted odds ratio of an early treatment response of 9.26, 95%CI = [3.78, 25.01] in favor of the carbapenem group, which corresponds to an average marginal effect of 32.4%, 95%CI = [22.7%, 42.0%]. Detailed information on the adjusted effect estimation can be found in the online Appendix A).

Treatment failure in the piperacillin/tazobactam group was mainly based on a numerically higher rate of patients with hemodynamic instability (45.7% versus 31.7%), increased SOFA score (38.3% versus 23.2%) and persistent or relapsed bacteremia (18.8% versus 7.4%) compared to the carbapenem group. For details please refer to Table 2.

As a consequence, piperacillin/tazobactam was escalated on median day 3 (interquartile range: day 1.8–5.0) in 30 patients (37.0%) to meropenem (*n* = 15), any combination (*n* = 14) or tigecycline (*n* = 1). Carbapenems were de-escalated on a median of day 4 (interquartile range: day 3-6) in 14 patients (17.1%) to fluoroquinolones (*n* = 7), piperacillin/tazobactam (*n* = 5) and cephalosporins (*n* = 2).

### 2.4. Secondary Outcomes

At day 14, the overall clinical success rate did not significantly differ between the empiric piperacillin/tazobactam group and the empiric carbapenem group (69.1% versus 63.4%, *p* = 0.617). Patients who received empiric piperacillin/tazobactam treatment had a numerically lower 14-day mortality rate than patients who received carbapenems as the initial antimicrobial treatment (4.9% versus 12.2%, *p* = 0.160).

In 18/180 patients (10.0%) with follow-up BCs, persistent or relapsed bacteremia with either *Enterobacter* spp. (*n* = 10), *Serratia* spp. (*n* = 6), *Citrobacter freundii* complex (*n* = 1) or *Morganella morganii* (*n* = 1) was detected. Among patients with follow-up BCs, persistent or relapsed bacteremia was found in 9/48 (18.8%) patients in the piperacillin/tazobactam group, 4/54 (7.4%) patients in the carbapenem group, 1/4 (25%) patients in the cotrimoxazole group, 1/29 (3.4%) patients in the fluoroquinolone group, 2/17 (11.8%) patients in the cephalosporin group, and 1/24 (4.2%) patients in the combination group. In vitro piperacillin/tazobactam and ceftazidime resistance were detected in 3 patients with relapsed *Enterobacter* bacteremia (at 23 days, 24 days and 47 days after the initial positive BC) who had received piperacillin/tazobactam for the initial BSI episode.

The main targeted treatments for SPICE-BSI included carbapenem monotherapy (*n* = 91), followed by fluoroquinolone monotherapy (*n* = 74), piperacillin/tazobactam monotherapy (*n* = 58), and targeted combination therapy (*n* = 38). Relapse SPICE-BSI rate in patients with follow-up BCs was 7.7% in the targeted piperacillin/tazobactam group, 3.1% in the targeted meropenem group, 2.8% in the targeted fluoroquinolone group and 0.0% in the targeted combination group. The clinical success rate at day 14 and 14-day survival rate were highest in the targeted fluoroquinolone group (78.4% and 97.3%) and lowest in the targeted combination group (55.3% and 84.2%).

### 2.5. Predictors of Early Treatment Response

As shown in Table 3, SOFA score at baseline (adjusted odds ratio (AOR): 0.83, 95% confidence interval (CI) 0.77–0.91), chronic liver disease (AOR 0.32, 95% CI 0.13–0.82) and empiric monotherapy with piperacillin/tazobactam (AOR 0.25, 95% CI 0.12–0.53) were identified as independent negative predictors and cholangitis (AOR 3.49, 95% CI 1.36–8.94) as an independent positive predictor of early treatment response in the multivariable model.

## 3. Discussion

Antimicrobial treatment of patients with SPICE-BSI remains challenging in clinical practice. Although carbapenems are the gold standard for treatment of severe infections, alternative carbapenem-sparing treatment options are needed [10,11,12,14] to prevent increases in carbapenem resistance [15]. Whereas most previous studies have assessed treatment outcomes by comparing 30-day mortality rates among different treatment regimens [10,11,16,17], this study focused on early treatment response on day 3 as the primary outcome. The main findings of our study were that early treatment response was significantly lower in the piperacillin/tazobactam group than in the carbapenem group (*p* = 0.006) despite similar disease severities (median SOFA score: 3.0 versus 4.5, *p* > 0.2) and that empiric piperacillin/tazobactam use (AOR 0.25, *p* < 0.001), baseline SOFA score (AOR 0.83, *p* < 0.001) and liver comorbidity (AOR 0.32, *p* = 0.018) were independently associated with early treatment failure. Additionally, the persistent or relapsed bacteremia rate was numerically higher in the piperacillin/tazobactam group (18.8% in patients with follow-up BCs and empiric piperacillin/tazobactam treatment versus 7.4% in patients with follow-up BCs and empiric carbapenem treatment). Development of resistance to the initial treatment regimen was only detected in a limited number of patients with relapsed *Enterobacter*-BSI who had received piperacillin/tazobactam (3/48 patients with follow-up BCs, 6.3%). However, compared to carbapenem monotherapy, empiric piperacillin/tazobactam treatment did not reduce overall clinical success on day 14 or 14-day mortality, most likely due to the high escalation rate (37.0%) in the piperacillin/tazobactam group. Similarly, a systematic review and meta-analysis of 13 observational studies including 1021 patients that compared beta-lactam/beta-lactamase inhibitors with carbapenems as definitive therapies for BSI with potential AmpC-producing Enterobacterales did not find significant differences in 30-day mortality (OR: 1.13, 95% CI: 0.58–2.20) [11].

In accordance with the findings of a recent retrospective cohort study including 241 patients with SPICE-BSI that was conducted at two university teaching hospitals in Singapore [16], *Enterobacter* spp. (52.5%) were the most common SPICE organisms isolated from our patients’ BCs, and carbapenems (28%) and piperacillin/tazobactam (27%) were the most common active empiric antibiotics used. In contrast, pneumonia (27.8%) was the most common source of bacteremia in the present study (23.5% in the empiric piperacillin/tazobactam group and 35.4% in the empiric carbapenem group), and patients in the empiric piperacillin/tazobactam group had the lowest rate of urinary tract infection (8.6%). Contrary, in the Singapore study by Tan et al., urinary tract infection and vascular catheters were the most common sources of bacteremia [16]. Recent studies comparing carbapenem and non-carbapenem therapy regimens for SPICE-BSI have mentioned adequate source control as a prerequisite for successful therapy with piperacillin/tazobactam [11,13]. In the present study, the piperacillin/tazobactam and meropenem group had similar rates of early source control. However, it has previously been theorized that the focus of infection cannot be detected properly in a relevant number of cases [16]. In fact, patients in the piperacillin/tazobactam group had a high rate of unknown focus, occurring in 24.7% of patients.

We identified SOFA score at admission, liver comorbidity and empiric use of piperacillin/tazobactam as independent risk factors for early clinical failure in patients with SPICE bacteremia. Associations of acute illness severity or liver comorbidity with clinical outcomes have been reported in many studies [16,18,19], but only a few studies have indicated that beta-lactam/beta-lactamase inhibitors are inferior to carbapenems [20,21]. A cohort study from New York comparing treatment outcomes of BSIs caused by *Enterobacter* spp., *Serratia* spp. or *Citrobacter* spp. between patients receiving piperacillin/tazobactam (*n* = 88) and patients receiving meropenem or cefepime (*n* = 77) concluded that piperacillin/tazobactam may be a valuable treatment option for SPICE-BSI, although persistent bacteremia at ≥72 h was also more common in the piperacillin/tazobactam-treated patients than in the meropenem- or cefepime-treated patients (8/41 versus 4/41 patients in the propensity score-matched cohort) [17]. In the present study, the daily piperacillin/tazobactam dose was mainly 13.5 g given as an intermittent bolus infusion in the normal ward and as a continuous infusion in the ICU. This dose is lower than that in an ongoing randomized controlled trial investigating the effects of piperacillin/tazobactam (4.5 g every 6 h i.v.) versus meropenem (1 g every 8 h i.v.) on BSIs caused by AmpC producers (https://clinicaltrials.gov/ct2/show/NCT02437045, accessed on 1 May 2021).

Our study has a few limitations. First, the study was a retrospective single-center study. Therefore, AmpC beta-lactamases were not routinely examined. Second, therapeutic drug monitoring was not routinely performed, even in patients receiving continuous infusion. Third, whether all repeated bacteremia episodes were true relapse events versus new infections remains unclear, as isolates were not available for molecular typing. Fourth, follow-up BCs were performed in only a subset of patients (61%). However, a recent review has concluded that under conditions of adequate source control and a lack of risk factors or concern for endovascular infection, most Enterobacterales-related BSIs do not require routine follow-up BC [22].

## 4. Materials and Methods

### 4.1. Study Population and Study Design

This retrospective single-center cohort study was conducted at the University Hospital of Jena, Germany, a 1400-bed academic hospital and the seat of the clinics of the Medical University of Jena. After approval by the ethics committee, we retrospectively included all hospitalized adult patients for whom SPICE organisms had been isolated from blood cultures (BCs) between January 2011 and February 2019. SPICE-BSI was defined as the presence of at least one positive BC for any SPICE organism (including one of the following: *Enterobacter* spp., *Serratia* spp., *Citrobacter freundii* complex, *Klebsiella aerogenes*, *Providencia* spp. and *Morganella morganii*) in patients with any suspicion of an infection (e.g., fever, hypothermia, elevated inflammatory parameters, or clinical signs of infection). The exclusion criteria were an age <18 years and in vitro resistance to empiric antibiotic treatment. In addition, patients who received <3 days of antimicrobial therapy because of palliative care, death within the first 48 h after BC or transfer to another hospital were excluded. After inclusion, the patients were assigned to different treatment groups according to their initial antibiotic regimens (carbapenem monotherapy, piperacillin/tazobactam monotherapy, cephalosporin monotherapy, fluoroquinolone monotherapy, cotrimoxazole monotherapy, gentamicin monotherapy, or any combination therapy). Only antimicrobial agents with Gram-negative activity were recorded. The choice of the empiric and targeted antibiotic treatment was at the discretion of the treating physicians and was based on international and institutional antibiotic guidelines. In our hospital, piperacillin/tazobactam monotherapy is the primary recommended empiric treatment in patients with sepsis and no documentation of prior infection or colonization with MRSA or multidrug-resistant Gram-negative bacteria. Meropenem is given in case of treatment failure or as a second choice.

### 4.2. Microbiological BSI Diagnostics

Each BC set consisted of a BD BACTEC Plus Aerobic/F and a Lytic/10 Anaerobic/F bottle (BD Diagnostics, Heidelberg, Germany). Usually, 1–3 BC sets were collected per blood draw. The BC bottles were incubated on a BACTEC FX instrument (BD Diagnostics) for up to 5 days. Gram staining and subculturing of positive BCs were performed using standard microbiological methods. Isolates were identified by Vitek MS (bioMérieux, Nürtingen, Germany). Antimicrobial susceptibility testing (AST) was performed using a Vitek 2 (bioMérieux), and the minimal inhibitory concentrations were determined according to the European Committee on Antimicrobial Susceptibility Testing (EUCAST) criteria. AmpC beta-lactamases were not examined via routine susceptibility testing. Phenotypic AST using Vitek 2 only included the identification of ESBL-mediated resistance mechanisms.

### 4.3. Data Collection

The following data were collected from the patients’ medical records: general demographic data (sex, age, and body mass index); comorbidities, including the Charlson Comorbidity Index; presence of severe immunodeficiency or implanted devices; source of SPICE-BSI (e.g., vascular catheter-related, respiratory, urinary, biliary, abdominal, surgical site, skin/soft tissue, or deep organ space infection); SPICE organism(s) isolated from the BC; admission to an ICU; hemodynamic stability, fever > 38 °C, leukocytosis >12 Gpt/l, Sequential Organ Failure Assessment (SOFA) score at baseline (day of BC), 72 h and 14 days after initial positive BC and start of active treatment; empiric and targeted antibiotic therapy; duration of antimicrobial treatment; source control; duration of bacteremia; length of hospital stay and discharge mode (deceased, transferred to another hospital, or discharged home).

### 4.4. Outcome Measures

The primary outcome of the study was early treatment response 72 h after the establishment of a positive BC and start of active antimicrobial treatment. Early treatment response required all of the following: survival, hemodynamic stability without inotropes, improved or stable SOFA score [23], resolution of fever (>38 °C), resolution of leukocytosis (white blood cell count >12 Gpt/l) [24] and microbiologic resolution (no documented relapse or persistent bacteremia).

The secondary outcomes included the clinical success 14 days after the initial positive BC, the 14-day mortality rate and relapse or persistent bacteremia. Clinical success was defined as a composite of patient survival, hemodynamic stability, an improved or stable SOFA score, resolution of fever and leukocytosis and no relapse bacteremia. Relapse or persistent bacteremia was defined on the basis of a positive BC for the same bacterial species collected beyond 72 h after the initial positive BC [13] until day 60.

### 4.5. Statistical Analysis

Data analysis was performed using SPSS software, version 27.0 (SPSS Inc., Chicago, IL, USA) and R, version 4.0.5 [25]. The patients’ demographics, comorbidities, disease severities and bacteremia sources were compared among the different empiric treatment regimens. Categorical variables were expressed as frequencies and percentages for the group from which they were derived. Continuous variables were expressed as medians together with the first and third quartiles (Q1, Q3). The baseline characteristics were compared among different empiric treatment groups including at least five patients using Fisher exact test for nominal data and the Kruskal–Wallis test for ordinal or numeric data. The Holm–Bonferroni method was used to adjust for multiple testing.

The primary outcome (early treatment response 3 days after the initial BC) and secondary outcomes (clinical success on day 14, 14-day mortality, persistent or relapsed bacteremia) were compared only between the groups receiving the two most common empiric treatments (piperacillin/tazobactam monotherapy versus carbapenem monotherapy) with Fisher exact tests using a two-sided *p*-value threshold of <0.05. The Holm–Bonferroni method was used to adjust for multiple testing. Covariate imbalance between the two treatment groups were assessed using absolute standardized mean differences (ASMD), adjusted odds ratios and average marginal effects were estimated based on multiple logistic regressions that included all covariates with ASMD ≥0.2.

To identify potential predictors of early clinical response, logistic regression analyses were performed. All variables associated with early clinical response with a *p*-value ≤ 0.1 in the univariate analysis were used to perform multivariable logistic regression analysis. For the multivariable model, a two-tailed *p*-value < 0.05 was considered to be significant.

## 5. Conclusions

In conclusion, our findings identify not only disease severity but also initial treatment with piperacillin/tazobactam, predominantly used at standard doses, as independent negative predictors of early treatment response in patients with BSI caused by potential AmpC beta-lactamase producing Enterobacterales where pneumonia was the main focus of bacteremia. In addition, we detected in vitro resistance to piperacillin/tazobactam in at least some patients with relapsed *Enterobacter*-BSI after empiric piperacillin/tazobactam treatment for the initial BSI episode. This might be of clinical relevance as, so far, EUCAST expert opinion recommendation only discourages the use of third-generation cephalosporin therapy for the treatment of potential AmpC-producing Enterobacterales infections (https://www.eucast.org/fileadmin/src/media/PDFs/EUCAST_files/Expert_Rules/2020/ExpertRules_V3.2_20190515_Enterobacterales.pdf, accessed on 27 May 2021). However, the causal interpretation of the reported treatment effects for piperacillin/tazobactam versus carbapenems is limited due to the retrospective study design. Randomized controlled trials are needed to constitute stronger empirical evidence of the superior effectiveness of the carbapenem regimen.

## Figures and Tables

**Figure 1 antibiotics-10-00665-f001:**
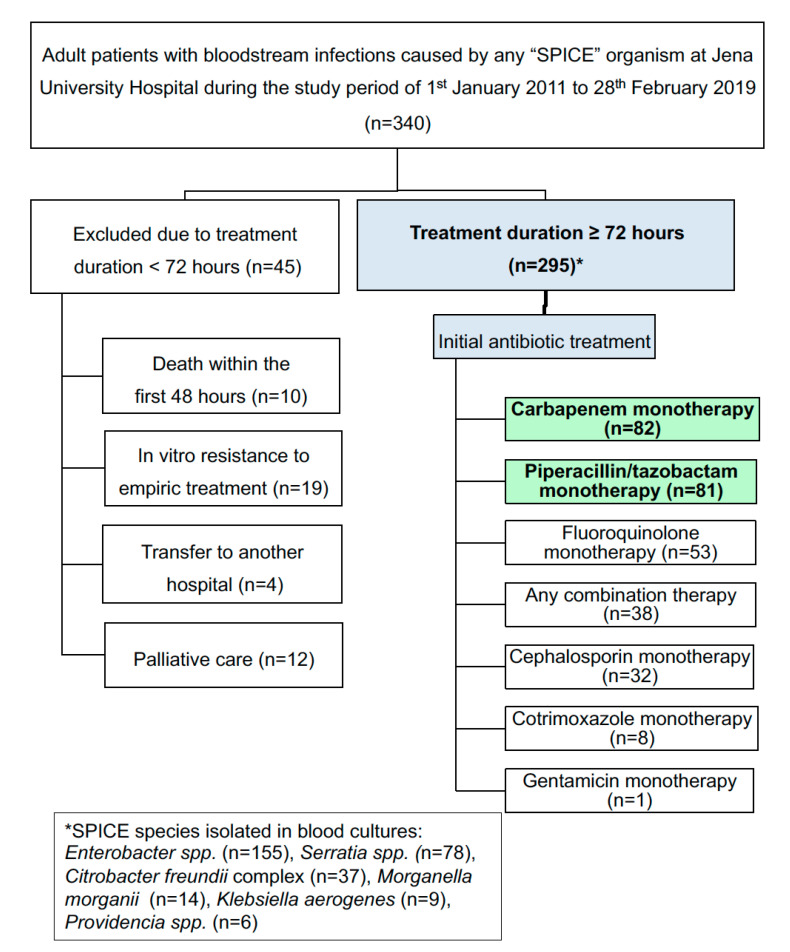
Flow chart of the study population and stratification of the included patients according to the initial antimicrobial treatment. Only patients with antimicrobial treatment for at least 3 days were considered for the study. Carbapenem monotherapy involved imipenem/cilastatin (*n* = 2) or meropenem (*n* = 80). Cephalosporin monotherapy involved cefuroxime (*n* = 2), ceftriaxone (*n* = 19), cefotaxime (*n* = 1) or ceftazidime (*n* = 10). Fluoroquinolone monotherapy involved ciprofloxacin (*n* = 45) or moxifloxacin (*n* = 8). Combination therapy involved piperacillin/tazobactam plus ciprofloxacin (*n* = 31), piperacillin/tazobactam plus ceftriaxone (*n* = 1), piperacillin/tazobactam plus gentamicin (*n* = 1), ceftazidime plus gentamicin (*n* = 1), gentamicin plus ciprofloxacin (*n* = 1), meropenem plus gentamicin (*n* = 1), meropenem plus fosfomycin (*n* = 1) or meropenem plus moxifloxacin (*n* = 1).

**Table 1 antibiotics-10-00665-t001:** Comparison of baseline characteristics, comorbidities, disease severities and bacteremia sources among patients with SPICE bloodstream infection (BSI) stratified according to their empiric treatment regimens (≥5 patients per group).

Variable	Empiric Treatment Groups						*p*-Value ^3^
	Monotherapy					Any Combi-nation (*n* = 38)	
	Cotrimoxazole (*n* = 8)	Fluoroquin-olone (*n* = 53)	Cephalosporin (*n* = 32)	Piperaci-llin/Tazo-bactam (*n* = 81)	Carbapenem (*n* = 82)		
Male sex	6 (75.0)	35 (66.0)	20 (62.5)	52 (64.2)	62 (75.6)	28 (73.7)	0.560
Age, years	50.5 (40.0–63.8)	68.0 (59.0–76.0)	68.0 (58.3–79.8)	68.0 (59.0–75.0)	66.5 (56.8–73.3)	68.0 (60.0–75.3)	0.893
Body mass index	24.8 (20.1–32.5)	25.7 (22.9–29.2)	25.0 (22.4–27.5)	26.1 (23.0–31.0)	26.0 (24.2–30.0)	26.2 (23.5–30.6)	0.374
Implanted device ^1^	2 (25.0)	10 (18.9)	2 (6.3)	13 (16.0)	21 (25.6)	20 (52.6)	0.002
Charlson Comorbidity Index	2.0 (1.0–3.0)	2.0 (1.0–4.0)	2.0 (0–6.0)	3.0 (2.0–4.0)	3.0 (2.0–5.0)	3.5 (1.8–5.0)	0.310
*Comorbidities* ^2^							
Heart failure	1 (12.5)	13 (24.5)	5 (15.6)	25 (30.9)	26 (31.7)	23 (60.5)	0.021
Lung disease	2 (25.0)	9 (17.0)	3 (9.4)	20 (24.7)	26 (31.7)	12 (31.6)	0.092
Kidney disease	0 (0.0)	3 (5.7)	2 (6.3)	16 (19.8)	11 (13.4)	8 (21.1)	0.087
Liver disease	0 (0.0)	12 (22.6)	3 (9.4)	13 (16.0)	14 (17.1)	3 (7.9)	0.342
Diabetes	1 (12.5)	11 (20.8)	8 (25.0)	18 (22.2)	24 (29.3)	15 (39.5)	0.334
Metastatic carcinoma /leukemia	1 (12.5)	2 (3.8)	4 (12.5)	8 (9.9)	4 (4.9)	5 (13.2)	0.266
Pitt bacteremia score	0 (0–4.3)	1.0 (0–2.0)	1.0 (0–1.3)	1.0 (0–2.0)	1.0 (0–4.0)	2.0 (0–7.0)	0.052
SOFA score at baseline	1.5 (0–6.0)	2.0 (1.0–6.5)	3.0 (1.0–6.5)	3.0 (1.0–7.5)	4.5 (1.0–11.0)	7.5 (3.0–11.3)	0.021
*Causative pathogen* ^2^							
*Enterobacter*	3 (37.5)	31 (58.5)	17 (53.1)	43 (53.1)	44 (53.7)	16 (42.1)	0.678
*Serratia*	3 (37.5)	12 (22.6)	6 (18.8)	18 (22.2)	26 (31.7)	13 (34.2)	0.410
*Other SPICE*	2 (25.0)	11 (20.8)	9 (28.1)	20 (24.7)	14 (17.1)	9 (23.7)	0.761
Polymicrobial BSI	2 (25.0)	11 (20.8)	6 (18.8)	22 (27.2)	13 (15.9)	5 (13.2)	0.441
*Main source of BSI* ^2^							
Unknown	3 (37.5)	10 (18.9)	8 (25.0)	20 (24.7)	16 (19.5)	8 (21.1)	0.790
Respiratory tract	0 (0.0)	7 (13.2)	4 (12.5)	19 (23.5)	29 (35.4)	22 (57.9)	<0.001
Urinary tract	3 (37.5)	13 (24.5)	8 (25.0)	7 (8.6)	16 (19.5)	5 (13.2)	0.900
Biliary tract	0 (0.0)	10 (18.9)	8 (25.0)	12 (14.8)	5 (6.1)	2 (5.3)	0.594
Vascular catheter	1 (12.5)	8 (15.1)	1 (3.1)	11 (13.6)	10 (12.2)	3 (7.9)	0.542
Intra-abdominal	1 (12.5)	3 (5.7)	1 (3.1)	4 (4.9)	6 (7.3)	1 (2.6)	0.749
Surgical site	0 (0.0)	2 (3.8)	2 (6.3)	9 (11.1)	10 (12.2)	4 (10.5)	0.571

The data are presented as the no. (%) or as the median (quartile 1–3). ^1^ Implanted devices include cardiac and orthopedic implanted devices. ^2^ More than one answer is possible. ^3^ To compare characteristics among patient subgroups, Fisher exact tests (for nominal data) or Kruskal-Wallis tests (for ordinal or numeric data) were used. The two-sided *p*-values are given. The provided *p*-values were corrected by the Holm–Bonferroni method for multiple testing.

**Table 2 antibiotics-10-00665-t002:** Comparison of SPICE bacteremia treatment outcomes between groups of patients treated with the two most common empiric regimens.

Variable	Piperacillin/Tazobactam Group (*n* = 81)	Carbapenem Group (*n* = 82)	*p*-Value ^4^
Standard dosage ^1^	78 (96.3)	75 (91.5)	0.211
vs.			
High dosage ^2^	3 (3.7)	7 (8.5)	
Admission to ICU	30 (37.0)	35 (42.7)	0.523
Treatment duration of initial regimen in days	5 (3–9)	8 (5.8–11)	0.021
Total treatment duration in days	11 (8–15)	10 (7–14)	0.374
Early treatment response day 3	17 (21.0)	40 (48.8)	0.006
Intensive care unit	2/30 (6.7)	7/35 (20.0)	0.161
Normal ward	15/51 (29.4)	33/47 (70.2)	0.002
Correlates of early treatment failure ^3^			
Early death until day 3	2 (2.5)	2 (2.4)	1
Hemodynamic instability	37 (45.7)	26 (31.7)	0.078
Any increase in SOFA score	31 (38.3)	19 (23.2)	0.061
Persistent fever >38 °C	16 (19.8)	13 (15.9)	0.419
Persistent leukocytosis	21 (25.9)	21 (25.6)	1
Persistent bacteremia ≥72 h or relapse bacteremia among patients with follow-up blood cultures (BCs)	9/48 (18.8)	4/54 (7.4)	0.136
Treatment escalation within 72 h	19 (23.5)	1 (1.2)	<0.001
Early source control ^3^	31 (38.3)	31 (37.8)	1
Vascular catheter removal	20 (24.7)	25 (30.5)	0.484
Urinary catheter removal	9 (11.1)	14 (17.1)	0.369
Implantation of biliary stent	6 (7.4)	0 (0)	0.28
Abscess drainage	2 (2.5)	2 (2.4)	1
Surgery	10 (12.3)	10 (12.2)	1
Clinical success day 14	56 (69.1)	52 (63.4)	0.617
14-day mortality rate	4 (4.9)	10 (12.2)	0.16
In vitro resistance to initial regime in patients with relapse bacteremia among patients with follow-up BCs	3/48 (6.3)	0/54 (0)	0.101

^1^ The standard piperacillin/tazobactam dosing regimen involved a 4.5 g bolus every 8 h in patients with creatinine clearance ≥20 mL/min/m^2^ (4.5 g every 12 h if < 20 mL/min/m^2^) in the normal ward or continuous infusion of approximately 13.5 g daily after an initial bolus of 4.5 g (only in the ICU). The standard carbapenem dosing regimen for meropenem or imipenem involved a 1 g bolus every 8 h in the normal ward or continuous infusion of 3 g daily after a bolus of 1 g (only in the ICU). ^2^ Piperacillin/tazobactam high dosage regimen was given as continuous infusion with a daily dose of 17–18 g after a bolus infusion of 4.5 g (only in the ICU). A high dose of meropenem was given as daily dose of 4 to 6 g either as continuous infusion after a bolus of 1–2 g or as intermittent infusion of 1g every 6 h up to 2 g every 8 h. The data are presented as the no. (%), the no./total no. (%), or the median (quartile 1–3). ^3^ More than one answer is possible. ^4^ To compare different treatment outcome parameters between the piperacillin/tazobactam and carbapenem groups, Fisher exact tests (for nominal data) and Mann–Whitney U tests (for ordinal or numeric data) were used. The two-sided *p*-values are given. The Holm–Bonferroni method was used to adjust for multiple testing.

**Table 3 antibiotics-10-00665-t003:** Predictors of early treatment response in patients with SPICE bacteremia.

Variable	Early Clinical Response		Odds Ratio (95% CI) *p*-Value	Adjusted Odds Ratio (95% CI) *p*-Value ^a^
	Yes (*n* = 119)	No (*n* = 176)		
Age, years	68.0 (60.0–76.0)	66.5 (56.5–74.8)	1.01 (0.99–1.03) *p* = 0.262	-
Male sex	84 (70.6)	120 (68.2)	1.12 (0.68–1.86) *p* = 0.661	-
Body mass index	25.9 (23.7–29.3)	25.9 (23.1–30.5)	1.00 (0.98–1.03) *p* = 0.696	-
Augmented renal clearance	6 (5.0)	8 (4.5)	1.13 (0.38–3.33) *p* = 0.832	-
SOFA score at baseline	2.0 (0–4.0)	6.0 (2.0–11.0)	0.80 (0.75–0.86) *p* < 0.001	0.83 (0.77–0.91) *p* < 0.001 *
ImpIanted devices	17 (14.3)	51 (29.0)	0.41 (0.23–0.76) *p* = 0.004	0.83 (0.38–1.84) *p* = 0.648
Severe immuno-deficiency	15 (12.6)	24 (13.6)	0.90 (0.45–1.79) *p* = 0.754	-
Chronic heart failure	30 (25.2)	64 (36.4)	0.59 (0.35–0.99) *p* = 0.045	0.86 (0.44–1.68) *p* = 0.662
Chronic respiratory disease	24 (20.2)	48 (27.3)	0.67 (0.39–1.18) *p* = 0.165	-
Chronic kidney insufficiency	5 (4.2)	35 (19.9)	0.18 (0.07–0.47) *p* < 0.001	0.50 (0.16–1.58) *p* = 0.239
Chronic liver disease	11 (9.2)	35 (19.9)	0.41 (0.20–0.85) *p* = 0.016	0.32 (0.13–0.82) *p* = 0.018 *
Diabetes mellitus	32 (26.9)	45 (25.6)	1.07 (0.63–1.82) *p* = 0.800	-
Metastatic carc-inoma /leukemia	10 (8.4)	14 (8.0)	1.04 (0.45–2.43) *p* = 0.924	-
Unknown focus	25 (21.0)	40 (22.7)	0.90 (0.51–1.59) *p* = 0.727	-
Pneumonia	17 (14.3)	65 (36.9)	0.29 (0.16–0.52) *p* < 0.001	0.51 (0.23–1.10) *p* = 0.080
Urinary tract infection	32 (26.9)	21 (11.9)	2.72 (1.48–5.00) *p* = 0.001	1.64 (0.74–3.62) *p* = 0.225
Cholangitis	23 (19.3)	14 (8.0)	2.77 (1.36–5.64) *p* = 0.005	3.49 (1.36–8.94) *p* = 0.009 *
Vascular catheter-related	17 (14.3)	17 (9.7)	1.56 (0.76–3.19) *p* = 0.225	-
Intra-abdominal infection	4 (3.4)	12 (6.8)	0.48 (0.15–1.51) *p* = 0.208	-
*Enterobacter* spp.	65 (54.6)	90 (51.1)	1.15 (0.72–1.83) *p* = 0.557	-
*Serratia* spp.	28 (23.5)	50 (28.4)	0.78 (0.45–1.33) *p* = 0.352	-
Other SPICE pathogens	27 (22.7)	38 (21.6)	1.07 (0.61–1.87) *p* = 0.823	-
Polymicrobial bacteremia	24 (20.2)	35 (19.9)	1.02 (0.57–1.82) *p* = 0.953	-
Empiric cephalosporin	17 (14.3)	15 (8.5)	1.79 (0.86–3.74) *p* = 0.122	-
Empiric cotrimoxazole	4 (3.4)	4 (2.3)	1.50 (0.37–6.10) *p* = 0.575	-
Empiric fluoroquinolone	26 (21.8)	27 (15.3)	1.54 (0.85–2.80) *p* = 0.155	-
Empiric piper-acillin/tazobactam	17 (14.2)	64 (36.2)	0.29 (0.16–0.53) *p* < 0.001	0.25 (0.12–0.53) *p* < 0.001 *
Empiric carbapenem	40 (33.6)	42 (23.9)	1.62 (0.97–2.70) *p* = 0.068	1.92 (0.96–3.84) *p* = 0.066
Empiric combination therapy	15 (12.6)	23 (13.1)	0.96 (0.48–1.93) *p* = 0.907	-
Early source control	55 (46.2)	56 (31.8)	1.84 (1.14–2.98) *p* = 0.013	1.15 (0.61–2.19) *p* = 0.668

The data are presented as the no. (%) or as the median (quartile 1–3). Augmented renal clearance = glomerular filtration rate >130 mL/min/m^2^. Implanted devices include cardiac and orthopedic implanted devices. Early source control includes vascular catheter removal, urinary catheter removal, implantation/exchange of biliary stent, abscess drainage and surgery. * Statistically significant in multivariable logistic regression analysis (*p*-value < 0.05).

## Data Availability

The datasets used and/or analyzed during the current study are available from the corresponding author on reasonable request.

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
