# Peer review of "Early Treatment Outcomes for Bloodstream Infections Caused by Potential AmpC Beta-Lactamase-Producing Enterobacterales with Focus on Piperacillin/Tazobactam: A Retrospective Cohort Study"

_antibiotics, 2021, doi:10.3390/antibiotics10060665_

Round 1
Reviewer 1 Report
This manuscript by Herrmann et al, the authors show that early treatment with piperacillin/tazobactam shows lower early treatment response compared to treatment with carbapenems for AmpC beta-lactamase producing Enterobacterales. This study addresses a common question amongst clinicians regarding the use of piperacillin/tazobactam against this group of SPICE organisms. The paper is well-written and sound in its approach. It would be helpful to know (perhaps in the table) which carbapenem(s) was/were used (I am assuming it was meropenem based on dosing information) and which cephalosporins and quinolones specifically were used. Otherwise, I think this is a very well-written manuscript with little need for much editing overall.Author Response
Response to Reviewer 1 Comments
Point 1:
This manuscript by Herrmann et al, the authors show that early treatment with piperacillin/tazobactam shows lower early treatment response compared to treatment with carbapenems for AmpC beta-lactamase producing Enterobacterales. This study addresses a common question amongst clinicians regarding the use of piperacillin/tazobactam against this group of SPICE organisms. The paper is well-written and sound in its approach. It would be helpful to know (perhaps in the table) which carbapenem(s) was/were used (I am assuming it was meropenem based on dosing information) and which cephalosporins and quinolones specifically were used. Otherwise, I think this is a very well-written manuscript with little need for much editing overall.
Response 1:
We thank the reviewer for the positive assessment and the recommendation to include the different antibiotics agents of carbapenems, cephalosporins and quinolones used. In the original manuscript we only included this information in the figure legend of Figure 1. In the revised version we now included the following in the main text.:
-Results line 85-56: Carbapenem monotherapy involved meropenem (n=80) and imipenem/cilastatin (n=2).
-line 89-91: The used fluoroquinolones were ciprofloxacin (n=45) and moxifloxacin (n=8). Cephalosporin monotherapy involved cefuroxime (n=2), ceftriaxone (n=19), cefotaxime (n=1) and ceftazidime (n=10).
In the footnote of table 2 the following was added:
-line 148-149: The standard carbapenem dosing regimen for meropenem or imipenem involved.
Reviewer 2 Report
Authors reported early treatment outcomes for bloodstream infections caused by potential AmpC beta-lactamase-producing Enterobacterales. The study is potentially of interest; however, there are important issues in methods that should be addressed and these important limits cannot be easily improved.
- First of all, it is not clear as AmpC beta-lactamases were identified. Authors shoudl report in methods and results microbiological identification.
- There is an important bias about empirical therapy. How was it categorized? What about the effect of targeted therapy?
- A propensity score matched analysis should be performed to weight the effect of confounders on outcome
In my opinion, the manuscript should be evaluated by a biomedical statistician before publication.
Author Response
Response to Reviewer 2 Comments
Point 1:
Authors reported early treatment outcomes for bloodstream infections caused by potential AmpC beta-lactamase-producing Enterobacterales. The study is potentially of interest; however, there are important issues in methods that should be addressed and these important limits cannot be easily improved.
Question: First of all, it is not clear as AmpC beta-lactamases were identified. Authors should report in methods and results microbiological identification.
Response 1:
We thank the reviewer for the recommendation to clarify this point. Our study is based on retrospective clinical data, as AmpC beta-lactamases were not examined via routine antibiotic susceptibility testing. According to EUCAST expert opinion the investigated Enterobacterales including Enterobacter spp., Serratia spp., Citrobacter freundii complex, Klebsiella aerogenes, Providencia spp. and Morganella morganii are potential AmpC-producers. Selection of AmpC de-repressed cephalosporin-resistant mutants may occur during monotherapy with a third generation cephalosporin or combination therapy with an aminoglycoside. The risk is relatively high in Enterobacter, K. aerogenes and Citrobacter and low in Morganella, Serratia and Providencia. https://www.eucast.org/fileadmin/src/media/PDFs/EUCAST_files/Expert_Rules/2020/ExpertRules_V3.2_20190515_Enterobacterales.pdf
The following was added in the main text:
-Material and Methods line 328-333: AmpC beta-lactamases were not examined via routine susceptibility testing. Phenotypic AST using Vitek 2 only included the identification of ESBL-mediated resistance mechanisms.
-Discussion line 285:
First, the study was a retrospective single-center study. Therefore, AmpC beta-lactamases were not routinely examined.
Point 2: There is an important bias about empirical therapy. How was it categorized?
Response 2:
The choice of the empiric and targeted antibiotic treatment was at the discretion of the treating physicians and was based on international and institutional antibiotic guidelines. In our hospital piperacillin/tazobactam monotherapy is the primary recommended empiric treatment in patients with sepsis and no documentation of prior infection or colonization with MRSA or multidrug-resistant gram-negative bacteria. Meropenem is given in case of treatment failure or as second choice. To point out differences between patients included in the different empiric regimens, we already compared demographics, comorbidities, severity of disease, bacterial species and primary source of bacteremia in Table 1 of the original manuscript. We agree with the reviewer and are aware that bias by indication is very common in observational studies. In order to assess the severity of covariate imbalance between the carbapenem and the piperacillin/tazobactam group, we now used absolute standardized mean differences (ASMD), which are commonly used in propensity score analyses. In additional analyses we adjusted for all covariates with ASMD ≥ 0.2 in multiple logistic regression models. Two models were estimated. Model A includes all fully observed covariates with ASMD ≥ 0.2, and Model B includes also covariates with missing values. Hence, Model A allows for adjusted effect estimation based on the full sample. Model B allows for more comprehensive adjustment but with a smaller sample (n = 157 instead of n = 163). We report the adjusted odds ratio of Model A in the paper as well as the marginal average effect, which has been estimated based on the parameter estimates of the logistic regression models. The adjusted effect estimates confirm the results of the crude differences in the early treatment response between the carbapenem and the piperacillin/tazobactam group. The adjusted odds ratio of the Carbapenem regimen compared to the piperacillin/tazobactam regimen was 9.26, 95%CI = [3.78, 25.01], in Model A and 10.41, 95%CI = [3.97, 30.59], in Model B, which is statistically significant (p < 0.001). The adjusted odds ratios for the carbapenem versus piperacillin/tazobactam regimen were even larger than the unadjusted odds ratio (3.59, 95%-CI = [1.83, 7.27]). The expected proportion of cases with early treatment response in Model A was 53.2% under the Carbapenem regimen versus 18.5% under the piperacillin/tazobactam regimen. The estimates of Model B were nearly identical (53.2% under the Carbapenem regimen, 19.1% under piperacillin/tazobactam regimen). The estimated average marginal effects (AVE) were 32.4%, 95%CI = [22.7%, 42.0%] for Model A, and 32.1%, 95%CI = [22.2%, 42.7%] for Model B. Detailed information about the covariate imbalance (i.e., tables with ASMDs), estimation of adjusted effects are presented in an additional online supplement.
The following was added:
-Results line 129-135:
Comparing both regimens under statistical control of covariates (i.e., baseline characteristics, comorbidities, disease severity and bacteremia sources) with absolute standardized mean differences ≥ 0.2, resulted in an adjusted odds ratio of an early treatment response of 9.26, 95%CI = [3.78, 25.01] in favor of the carbapenem group, which corresponds to an average marginal effect of 32.4%, 95%CI=[22.7%, 42.0%]. Detailed information on the adjusted effect estimation can be found in the online supplemental material.
-Material and Methods line 311-317:
The choice of the empiric and targeted antibiotic treatment was at the discretion of the treating physicians and was based on international and institutional antibiotic guidelines. In our hospital piperacillin/tazobactam monotherapy is the primary recommended empiric treatment in patients with sepsis and no documentation of prior infection or colonization with MRSA or multidrug-resistant gram-negative bacteria. Meropenem is given in case of treatment failure or as second choice.
-Material and Methods line 374-377:
Covariate imbalance between the two treatment groups were assessed using absolute standardized mean differences (ASMD), Adjusted odds ratios and average marginal effects were estimated based on multiple logistic regressions that included all covariates with ASMD ≥ 0.2
Point 3: What about the effect of targeted therapy?
Response 3:
We thank the reviewer for this important comment.
The following was added:
-Results line 186-193:
The main targeted treatments for SPICE-BSI included carbapenem monotherapy (n=91), followed by fluoroquinolone monotherapy (n=74), piperacillin/tazobactam monotherapy (n=58), and targeted combination therapy (n=38). Relapse SPICE-BSI rate in patients with follow-up BCs was 7.7% in the targeted piperacillin/tazobactam group, 3.1% in the targeted meropenem group, 2.8% in the targeted fluoroquinolone group and 0.0% in the targeted combination group. Clinical success rate at day 14 and 14-day survival rate were highest in the targeted fluoroquinolone group (78.4% and 97.3%) and lowest in the targeted combination group (55.3% and 84.2%).
Point 4: A propensity score matched analysis should be performed to weight the effect of confounders on outcome.
Response 4:
Answer: Due to the limited number of cases for the comparison of carbapenem versus piperacillin/tazobactam treatment (n=163), we decided not to perform a propensity score matched analysis. According to a simulation study of Brookhart et al. (Brookhart MA, Schneeweiss S, Rothman KJ, Glynn RJ, Avorn J, Stürmer T. Variable selection for propensity score models. Am J Epidemiol. 2006;163(12):1149-56) on variable selection for propensity score (PS) building, the authors observed that the inclusion of variables that are strongly related to the exposure (i.e. the therapies to be compared) but only weakly related to the outcome can be detrimental to the precision of the estimate for the case of small studies (n=500 in their case). Furthermore, effect estimates obtained by propensity score matching are typically the average effect of the treated (ATT), sometimes the average effect of the untreated (ATU), but rarely the average treatment effect (ATE). Which of these effect estimates can be estimated by propensity score matching depends on the covariate distributions across treatment/control groups and the common support. In our study, there is no control group. Rather, we compare two alternative treatments. The most appropriate effect estimate in this case is the ATE, with one of the treatment groups as the arbitrarily chosen reference group. This effect can be estimated by means of a multiple logistic regression model, which includes the potential confounders as covariates alongside with the treatment variable. As outline in the response to the first question/comment, we reported adjusted odds ratios and marginal average effects in the paper and provide detailed information about covariate selection and balancing as well as the adjusted effect estimation in an online supplement. We also adapted the conclusion part.
-Conclusion line 393-399: This might be of clinical relevance as so far EUCAST expert opinion recommendation only discourages the use of third generation cephalosporin therapy for treatment of potential AmpC-producing Enterobacterales infections (https://www.eucast.org/fileadmin/src/media/PDFs/EUCAST_files/Expert_Rules/2020/ExpertRules_V3.2_20190515_Enterobacterales.pdf).
However, the causal interpretation of the reported treatment effects for piperacillin/tazobactam versus carbapenems is limited due to the retrospective study design. Randomized controlled trials are needed to constitute stronger empirical evidence of superior effectiveness of the carbapenem regimen.
Point 5: In my opinion, the manuscript should be evaluated by a biomedical statistician before publication.
Response 5:
We followed your recommendation and included Dr. Norman Rose, a biomedical statistician as coauthor.
Reviewer 3 Report
This manuscript by Herrmann reported outcomes for BSI caused by chromosomally-AmpC producing Enterobacterales with antibiotics treatment. The authors concluded that early treatment response rate on day 3 as the primary outcome was significantly lower in the PIP/TAZ treatment group than in the carbapenem treatment group. On the other hand, empiric PIP/TAZ treatment did not affect overall clinical success on day 14 or 14 day mortality, probably due to the high escalation rate in the PIP/TAZ group. The authors evaluated the outcomes of antibiotics treatment with focus on early response rate, which are thought to be novel, considering that most previous studies employed 30-day mortality rates. Overall, the manuscript was clearly written and the methodology employed was straightforward. I have some primarily minor comments to improve the manuscript.
- Introduction: Please delete "plasmid-encoded". SPICE should be handled as chromosomally AmpC-producer.
- Bacteria: As you know, Citrobacter species include non-chromosomally AmpC-producer such as Citrobacter koseri. Did the authors exclude them from the study? In addition, Enterobacter aerogenes is now handled as Klebsiella aerogenes. The description for bacterial species should be written in datil as much as possible.
Author Response
Response to Reviewer 3 Comments
Point 1:
This manuscript by Herrmann reported outcomes for BSI caused by chromosomally-AmpC producing Enterobacterales with antibiotics treatment. The authors concluded that early treatment response rate on day 3 as the primary outcome was significantly lower in the PIP/TAZ treatment group than in the carbapenem treatment group. On the other hand, empiric PIP/TAZ treatment did not affect overall clinical success on day 14 or 14 day mortality, probably due to the high escalation rate in the PIP/TAZ group. The authors evaluated the outcomes of antibiotics treatment with focus on early response rate, which are thought to be novel, considering that most previous studies employed 30-day mortality rates. Overall, the manuscript was clearly written and the methodology employed was straightforward. I have some primarily minor comments to improve the manuscript.
Question: 1. Introduction: Please delete "plasmid-encoded". SPICE should be handled as chromosomally AmpC-producer.
Response 1:
As recommended, we deleted plasmid-encoded in the abstract and the introduction.
Abstract Line 20-21 and Introduction Line 40-41: Enterobacterales that may harbor inducible chromosomal AmpC beta-lactamases genes.
Point 2:
Question: 2. Bacteria: As you know, Citrobacter species include non-chromosomally AmpC-producer such as Citrobacter koseri. Did the authors exclude them from the study? In addition, Enterobacter aerogenes is now handled as Klebsiella aerogenes. The description for bacterial species should be written in datil as much as possible.
Response 2:
We are grateful for this important comment. In the revised manuscript we only included Citrobacter freundii complex and did not consider two patients with Citrobacter koseri who were included in the original manuscript. Additionally, Klebsiella aerogenes were deleted from the Enterobacter group.
For detailed description of bacterial species see Results line 64-69: Enterobacter spp. (n=155) were the most common SPICE pathogens, followed by Serratia spp. (n=78), Citrobacter freundii complex (n=37), Morganella morganii (n=14), Klebsiella aerogenes (n=9) and Providencia spp. (n=6). Enterobacter spp. included Enterobacter cloacae complex (n=154) and Enterobacter cancerogenus (n=1), Serratia spp. included Serratia marcescens (n=75), Serratia liquefaciens (n=2) and Serratia odorifera (n=1), Providencia spp. included Providencia rettgeri (n=3) and Providencia stuartii (n=3).
Round 2
Reviewer 2 Report
Authors improved manuscript. I don't have further comments.